# Machine Learning-Driven Multiobjective Optimization: An Opportunity of Microfluidic Platforms Applied in Cancer Research

**DOI:** 10.3390/cells11050905

**Published:** 2022-03-05

**Authors:** Yi Liu, Sijing Li, Yaling Liu

**Affiliations:** 1School of Engineering, Dali University, Dali 671000, China; lyly1974@hotmail.com; 2Department of Bioengineering, Lehigh University, Bethlehem, PA 18015, USA; 3Department of Mechanical Engineering and Mechanics, Lehigh University, Bethlehem, PA 18015, USA

**Keywords:** cancer, cell sorting, circulating tumor cells, microfluidics, machine-learning

## Abstract

Cancer metastasis is one of the primary reasons for cancer-related fatalities. Despite the achievements of cancer research with microfluidic platforms, understanding the interplay of multiple factors when it comes to cancer cells is still a great challenge. Crosstalk and causality of different factors in pathogenesis are two important areas in need of further research. With the assistance of machine learning, microfluidic platforms can reach a higher level of detection and classification of cancer metastasis. This article reviews the development history of microfluidics used for cancer research and summarizes how the utilization of machine learning benefits cancer studies, particularly in biomarker detection, wherein causality analysis is useful. To optimize microfluidic platforms, researchers are encouraged to use causality analysis when detecting biomarkers, analyzing tumor microenvironments, choosing materials, and designing structures.

## 1. Introduction

Cancer includes sustaining proliferative signaling, evading growth suppressors, resisting cell death, enabling replicative immortality, inducing angiogenesis, activating invasion, reprogramming energy metabolism, evading immune destruction, building tumor microenvironment, and metastasis [1]. Cancer metastasis is a key contributor to cancer incidence and fatality. Circulating tumor cells (CTCs) can serve as a primary indicator of metastasis [2]. The isolation, identification, and characterization of CTCs are meaningful for metastasis research [3]. Traditional approaches to cancer metastasis are limited in physiological relevance in microenvironments (particularity with invasion assays based on Petri dishes), lack quantification accuracy with trans-well-based transmigration assays, and have challenges in spatiotemporal control in animal experiments. To overcome these limitations, microfluidic platforms can be developed to study the process of cancer metastasis in real-time quantitative analyses of physiological conditions.

Metastasis has been investigated comprehensively—biologically, mechanically, chemically, and physically. Because of the impediments of various disciplines, the causality of the substantial data has not been evaluated. With the development of data science, machine learning (ML) can be used as a powerful tool for cancer research [4,5,6]. ML draws from the large amount of data generated from microfluidic platforms themselves to assist in different assignments such as feature extraction, classification, prediction, and optimization. The microfluidics of intelligent algorithms have shown its capability to settle tough or even impossible problems, such as the optimization of microfluidic platform design, which traditional data analysis cannot achieve [7]. The causality of the different factors of distinct properties is key in pushing cancer metastasis research to a higher level. The integration of ML algorithms can reveal the causality of factors in the mechanisms of cancer and cancer metastasis.

## 2. Systematic Description

Cancer metastasis is a disease in which tissue cells grow uncontrollably and escape from the position of the primary tumor to invade other distant tissues or organs via the circulatory system of the body (see Figure 1), a process that substantially boosts the morbidity and lethality of cancer in patients. Cancer can easily go unnoticed at early stages because of a lack of clinical symptoms and because histopathological examination is not useful in detecting the primary extension of tumor cells in general [8]. Tumor cells become circulating tumor cells (CTCs) when they intrusively enter the circulation system that causes cancer metastasis. There are only 0.001% to 0.02% tumor cells associated with cancer metastasis. However, they contribute to over 90% of cancer-related deaths because they successfully metastasize [9]. Cancer metastasis triggers systematic pathological changes that need whole-body treatment, including surgical operation and chemotherapy due to lesions in multiple secondary organs. Diverse tumors metastasize specifically to particular secondary organs or tissues due to distinctive signals that stimulate the interaction between cell types and respond to the mechanical properties of microenvironments or the chemokines of secondary tumor sites [2,8,9]. In the process of metastasis, the early diagnosis and prognosis of cancer patients largely depends on the isolation and analysis of CTCs. The early diagnosis of CTCs can offer a promising proposal of treatment to cancer patients who need either further systemic treatment or targeted therapy after the initial tumor removal. Analyzing CTCs, mimicking tumor microenvironments, and investigating how tumor cells react to specific signals in vitro can lead to knowledge of the mechanisms in cancer metastasis and provide suggestions to prevent metastasis with both therapeutic drugs and therapeutic devices.

Intelligent microfluidics is an emerging field that integrates ML with devices. Microfluidics is a technology characterized by sub-millimeter structures and fluid control at the micrometer scale with extreme sensitivity and satisfied throughput. It has been used in improving diagnostics and biology research. Most of its applications can be subsumed under five broad categories: the analysis of DNA and proteins [10], the sorting of cells [11], high-throughput screening, chemical reactions [12,13], and transferring small volumes of materials [11]. The advancement of micro-/nanofabrication techniques and rapid prototyping enables the design of multifunctional microfluidics to investigate the isolation of CTCs and metastatic microenvironments. Microfluidic techniques are gradually becoming a powerful means of cancer metastasis research in fundamental and applied stages, and they are naturally associated with simple operation, cost efficiency, and the accurate manipulation of fluid volume. On the one hand, despite substantial progress, it is still a challenge to fuse biochemical factors secreted by cells of the extracellular matrix in the tumor microenvironment (TME) [6] to confront physical factors such as matrix stiffness, shear stress, interstitial flow, topography, engineering design, and clinical requirements, and to customize these factors methodically into one microfluidic platform to reach its full potential. On the other hand, ML is a super-efficient data-processing and -analysis method that can generate information and rules from large datasets and then re-program based on these analyses [7]. ML has been widely used in biological applications, such as labeled cell classification with biological immuno-properties, the label-free detection of cells with physical properties, and the opposite design of microfluidic platforms [7,14]. This innovative combination of two techniques, intelligent microfluidics, not only implements the dominant conveniences of microfluidics, but also uses the strength of ML to help microfluidic platforms live up to their enormous potential. Microfluidic platforms applied in cancer research have two major applications: CTC isolation [3,8,9] and tumor modeling to mimic metastatic microenvironments.

These microfluidic platforms can be divided into two main categories, “affinity-based strategies” and “label-free strategies”. The categories correspond to particular tumor markers and specific biophysical properties, respectively. The CTCs’ isolation of affinity-based strategies refers to the affinity of exact antigens conveyed on the membrane of the cell to bind with corresponding antibodies coated on the surface of microfluidic platforms or magnetic beads [2]. These tactics either separate CTCs by targeting epithelial cell adhesion molecule (*EpCAM*), epidermal growth factor receptor (*EGFR*), and human epidermal growth factor receptor2 (*HER2*) for positive selection, or by eliminating hemocytes by targeting distinct antigens such as leukocyte common antigen (*CD45*) via negative selection [2,15,16] (see Figure 2a). The principle of these methods is that antigen- and antibody-related cells adhere to the microfluidic platform surface with a high purity, whereas the majority of other cells are swept along with the fluid flow (see Figure 2).

Label-free-based strategies stand for CTC separation by biophysical properties other than the distinctive biomarkers on the cancer cells’ surface and the nucleic acid inside the cytoplasm [3,17,18,19]. CTCs are isolated from blood cells by their biophysical properties, such as cell morphology, buoyant density, electric charge, and deformability (see Figure 2b), to facilitate subsequent downstream analysis in cancer metastasis [18,19,20,21]. Label-free-based strategies of CTC isolation based on physical characteristics include four main types: density gradient centrifugation methods [22], filtration-based methods [23,24], hydrodynamic-based methods [15,25], and dielectrophoresis-based methods [26,27]. Density gradient centrifugation methods utilize differences in cell migration distances based on differences in cell buoyant density to isolate CTCs from background cells such as erythrocytes and leukocytes. The filtration-based method, a means that features uniform microstructures in filters, separates CTCs independently and simply according to the diversity of cell sizes and the deformability of different phenotypes with high throughput. A commercialized platform called “isolation by size of epithelial tumor cells (ISET)” has been demonstrated by Sun et al. to isolate CTCs in an inexpensive and efficient way [24]. The principle of hydrodynamic-based methods is the application of inertial force generated by fluid shear within micro-channels [25]. The flowing particles are migrated into different equilibrium positions by lateral force and hydrodynamic inertial effects based on their diversity in density and size. Electrophysical properties are applied for the dielectrophoresis-based method to isolate targeted CTCs under electric fields without regularity. Differences in inherent size and dielectric constants between healthy hemocytes and cancer cells produce different dielectrophoretic forces. The dielectrophoretic force is one of the bioparticle’s active manipulation forces, which manipulates and controls the motion of cells in microfluidic platforms with an irregular electric field due to the phenomenon of polarization [27,28]. It is an effective method applied for CTCs’ classification [27].

Above all, the microfluidics of CTC isolation, whether affinity or label-free, can achieve good efficiency and throughput. However, every method has its own drawbacks that influence isolation performance, such as the purity and recovery rate of CTCs. For the affinity-based strategies mentioned above, microfluidic fabrication and surface microstructure are crucial for the performance of microfluidic platforms. Microfluidic fabrication varies in materials including metal, silica, biopolymer, quantum dots, iron oxide, carbon-based, rare-earth-based, and other nanomaterials. Some particular biomedical applications have requirements for material composition, surface modification, and specific morphology. Polydimethylsiloxane (PDMS) is one of the most popular materials used for microfluidics because of its biological compatibility, low cost, express prototyping, gas permeability, optical properties, and mechanical properties such as elasticity and flexibility [29]. To increase the quantity of target cell capturing, superficial expansion is the solution, and a herringbone pattern is suggested to such microfluidic platforms [2,30]. It is imperative that a consolidation of the respective advantages of multiple methods is integrated into a microfluidic platform of acceptable capture capability. Label-free strategies can be used as the first step to eliminate leukocytes, followed by affinity-based methods to isolate CTCs with high purity and efficiency. The integration of choosing materials, the surface microstructures of microfluidics, and biochemical/biophysical response is typically based on the limited experience and knowledge of infinite data and causality. Materials, structures, and biochemical/biophysical responses are three essential elements of microfluidic chips. To reach maximal efficacy, the cross-talk or the causality of these three factors should be investigated. With the advancement of ML in both microfluidics and manufacturing, it is reasonable to expect that the combination of diverse classification methods could achieve CTCs’ separation automatically with higher efficiency, purity, and throughput.

In contrast to traditional statistic-based data analysis, ML features the dynamic advancement of predication aided by computers that can deal with large amounts of data, decreasing human intervention and workload. As mentioned above, over 50% of the applications of intelligent microfluidics in biotechnology include cell classification, cell screening, cell sorting, the identification of cellular pathology, and the measurement of cellular chemistry [3,18,21]. ML can also be used in the analysis, design, and manipulation of continuous or separate fluids in micro-constructs in order to optimize microfluidic platforms [14,17,31,32,33].

ML algorithms such as convolutional neural networks (CNNs), logistic regression (LR), support vector machines (SVMs), linear regression, variational autoencoders (VAEs), etc., are applied to cell classification based on microfluidic flow control, cell morphology, DNA quality, target antibodies, and cell density, either simultaneously or respectively.

In the cytopathologic analysis of cell classification, ML algorithm linear discriminant analysis (LDA) is used for the identification of pancreatic carcinoma cells within track-traced magnetic nano-pore microfluidics [34]. Quadratic discriminant analysis is used in the classification and differentiation of dissimilar types of label-free tumor cells [35]. LR-based linear classifiers use microfluidic imaging and analysis to capture and isolate cancer-related fibroblasts and two different cell types of lung cancer [36]. A decision tree can help classify label-free cancer cells in blood with continuous flow in microfluidic channels.

In cellular chemical analysis, SVM can be used for drug-sensitive and -insensitive tumor cell identification without labeled biomarkers [37]. CNN is applied in label-free cancer cells survival tests for drug bright-field discrimination on a large scale [38]. Manak et al. reported that random forest (RF) is used to analyze cell cultures of both solid prostatic carcinoma tissue and breast cancer tissue with a high-content biomarker assay on the surface of microfluidics owing to single-cell resolution [39]. Intelligent microfluidics can utilize a self-organizing map to identify blood samples based on the movement predication of erythrocytes. It can also use auto-encoding to evaluate the drug sensitivity of leukocytes.

There are five main applications that use microfluidic ML in biotechnology, including cell classification, signal processing, DNA sequencing, flow sculpting, and cell segmentation. ML algorithms are applied to microfluidic platforms with different formats. They can be classified into three major types: droplet microfluidics (see Figure 2b), trajectory prediction [40,41], and blood cell counting. Implementing ML in the format of droplet microfluidics is the most popular of these.

The cooperation of droplet microfluidics with a high-throughput nature and ML with good analysis efficiency can help improve the power of droplet-based microfluidic technology. Droplet microfluidic ML platforms have been restricted to the analysis of real-time or post-experiment data because of the limitation of standardized and enormous data sets [31]. The performance of droplet generators can be predicted by manipulating the design parameters. This ability omits expensive iterations of design and enables the optimization of application-specific design. Lashkaripouret et al. capitalized on using ML algorithms in a standardized large data set to develop DAFD (Design Automation of Fluid Dynamics), a kit on the Internet that can monitor and predict the work efficiency of droplet generators. Performance prediction provides information on additional characteristics such as estimating the influence of fabrication and testing tolerances with the observed performance. It can provide a criterion to regulate fluid with different flow rates to meet possible tolerances and be extensively used to encourage additional fluid combinations either by using a neural optimizer or through the transfer of learning [33]. In addition, Sarkar et al. demonstrated a semi-automated, droplet-based microfluidic platform that used ML to quantitatively compare the NK-92 cell interaction with different target cells used in cytotoxicity assay in immunotherapeutic conditions in antibody-based cancer [5]. The ML algorithm CNN and its algorithm implementation have been introduced to evaluate the dynamics of individual, effector-target cell pair conjugation and to target death in droplets. The study, based on the mass proteomic analysis of SKOV3 and SKBR3, revealed distinct changes in the stimulation of upstream regulators and cytotoxicity mediators and transcription, which was used to adjust the particular functionalities of NK-92 cells [5].

ML is applied to microfluidics based on the different properties (see Figure 3) of cells, such as bio-mechanical properties, optical properties, and electrical properties.

When detecting the mechanical properties of a cell, the integration of microfluidics technologies and ML can investigate cell behavior on a large scale through quantitative analysis to provide intelligent decisions and support clinical diagnostics. In order to remove red blood cells in advance in rare hereditary hemolytic anemia, a microfluidic ML platform has been designed to generate the physiological filtering function of a spleen to monitor blood disease based on the analysis of blood cell shapes in vitro [42]. It evaluates the deformability of red blood cells by squeezing them in planar orientation, while visually observing and calculating the capacity of red blood cells to reinstate their pristine shape after penetrating through constrictions such as microchannels or nano-pores. In this way, ML algorithms can distinguish abnormal red blood cells from normal ones. This microfluidic ML platform demonstrates the capability of recognizing and distinguishing between healthy controls and generic anemia patients, or rare hereditary hemolytic anemia subtypes with an average validity of 91% and 82%, respectively [42].

Integrating optical imaging techniques with microfluidics and ML allows the observation and extraction of information from biological or chemical samples. A deep-learning-based feature-fusion algorithm has been proposed by Hervé and his team to extract cell information from low-resolution lens-free cell images [43]. It improved not only the average ratio of information entropy by 7.69%, but also increased the standard deviation of images by 22.2%, leading to the efficient integration of cellular features. High-throughput single-cell analysis is a current challenge and a trend in cell classification research. Microfluidic Raman spectroscopy allows the isolation of a single cell, culturing in microenvironments, and simultaneous monitoring of cells in situ [44]. It has been proved to be successful in cell sorting, intra/extracellular variability, metabolic response to the environment, and the exploration of antibiotic susceptibility. The performance of this technique can be further improved by a combination of surface-enhanced Raman scattering (SERS) and microfluidics. SERS-based microfluidics generates advantages of better reproducibility, based on automation from microfluidics and the ultra-detection of SERS, and it is widely used in label-free detection, from carboxy-fluorescent-labelled DNA and miRNA to proteins such as IgG and bovine serum albumin, in the biological field [44,45,46]. The multiplex analysis of complex Raman spectra requires chemometric-based ML to perform a deconvolution analysis of output spectra in an efficient manner, or to apply SERS tags on the antigens of the cell’s surface, to complete an affinity-based cell detection. Wilson et al. utilized magnetic multicolor SERS nanotags to detect four breast cancer-related antigens on CTCs and original cancer cells with an immunomagnetic microfluidic platform to prove the reliability of multiple surface-marker detection on CTCs with their new method [16]. In these optical- and image-based detections, without adopting the information of high-resolution images, ML has the most appropriate strength for providing assistance to the logical and intelligent selection of rare cells, which are seldom known or have no unique biomarkers. An image-guided flow cytometer cell sorter based on ML offers a neoteric paradigm to support researchers and clinicians to separate cells with various user-defined features that comprise morphological/spatial peculiarity and fluorescent signals [17]. Gu et al. provided a microfluidic platform and a spatial–temporal transformation method to obtain real-time cell images using an extremely generic hardware. Supervised ML was utilized to create a methodology of user interface to produce sorting standards [47].

The biophysical properties of cell phenotype include cell size, shape, deformability, and subcellular structure, such as organelles in cytoplasm, mitochondrial network, membrane composition and morphology, ion pumps in endoplasmic reticulum, nucleus size, and so on. Fluorescence cytometry is not suitable to detect biophysical phenotypes, but it is highly sensitive to biochemical phenotypes. Nevertheless, impedance cytometry can assess the biophysical phenotypes of single cells with appropriate dielectric models, thereby making the quantification and stratification of complex samples in a label-free, multi-parametric manner possible [48]. The impedance cytometry data of cellular and subcellular features can influence the related subjects of microfluidic design, the applications of phenotype, and the analysis of data, including signal processing, dielectric modelling, and population clustering. For example, a recurrent neural network (RNN) has been used to enhance the details of cell characteristics by collecting the electric current signal through an electrical impedance flow cytometer of microfluidics [49]. It can theoretically process 2500 cells per second. Integrating impedance properties with microfluidic separation, sensor technique, and ML can create an efficient platform for the label-free quantification and isolation of subpopulations to arrange heterogeneous biosystems into strata.

In summary, intelligent microfluidics has supported plenty of applications in biotechnology, including cell classification, with the distinct properties of biochemistry, biophysics, and biology. In certain intelligent systems, the training data of the network has been generated entirely by the microfluidic platform itself. Despite the limitations of self-collected data, they can reach a promising outcome with advanced training through a network of previous training processes based on a transfer-learning discipline [33]. When it comes to images, CNN is the most popular algorithm of cell classification, owing to its extraordinary performance in deep learning. There are still many algorithms, such as SVM, LR, VAE, decision tree, LDA, PCA, and quadratic discriminant analysis, that are used in designing intelligent microfluidic systems of cell classification.

Moreover, with CNN, ML can perform cytotoxicity analysis, which is involved in the identification of cell-killing events caused by specific cells or distinct chemicals [5]. Cell trajectory predictions based on a self-organizing map model are decisive factors in the movement of blood cells in microchannels of microfluidics [40]. The monitoring of immuno-conditions based on cellular immunoassays with the support of CNN is time-efficient, highly parallel, and precise [50]. Multilayer perceptron (MLP) can construct bi-directional interfaces between artificial microfluidic platforms and biological tissues [32].

Intelligent microfluidic platforms can also be applied in areas of cancer screening based on biomarker detection. For example, various protein biomarkers and mRNA and miRNA markers [51] can all be utilized for the diagnosis of pancreatic cancers to achieve better particularity and sensitivity by minimizing the deviations of measurement from only one biomarker analysis. SERS-based ML can be applied in complex protein biomarker detection [4,16]. Meanwhile, key biomarker detection based on the integration of a classification tree and a K-nearest neighbor algorithm [52] can predict and differentiate distinct cancers with homogeneous biomarkers, as well as distinguish healthy individuals from cancer patients. Based on the accomplishments of key biomarker detections, a decision tree algorithm can predict different cancer types [52]. Similarly, there is something promising about single-cell resolution that Manak et al. revealed in the capability of random forest (RF) in risk stratification predictions for cancer patients based on phenotypic-biomarker assays of a live primary cell [39].

ML offers the opportunity to uncover the causality of different properties that are not easily detected by human experience. By taking advantage of different ML algorithms and distinct properties, there can be a better understanding of cancer biology and the multiplex mechanisms of drug tests. It is of great benefit to develop microfluidics to analyze tumor cell behavior, cancer metastasis, and drug screenings of different types of cancer.

## 3. Conclusions

Cancer metastasis is a complicated process. Microfluidic systems have been used for the diagnosis and testing of cancer metastasis. ML has been utilized in various ways, from the design of microfluidics, to data analysis based on different properties, to the phenotyping of cell populations [49]. One of the primary challenges in applying ML to cancer research is deciding on the most appropriate data types for training. Many factors contribute to validating the detection and testing of biochemical properties (specific protein expressions on the cell surface), biophysical properties (cell size, cell density, and cell morphology, and electromagnetism), and engineering characteristics (material, structure, surface property) [53]. However, not every factor is crucial in generating input training data. It is essential to find the causality among distinct factors for improving the diagnosis and testing of cancer metastasis.

In short, the combination of microfluidic platforms and ML may lead to innovative ways to isolate target cancer cells or to perform drug tests in tumor microenvironments. The causality analysis of ML may offer new working principles for the microfluidics industry.

## Figures and Tables

**Figure 1 cells-11-00905-f001:**
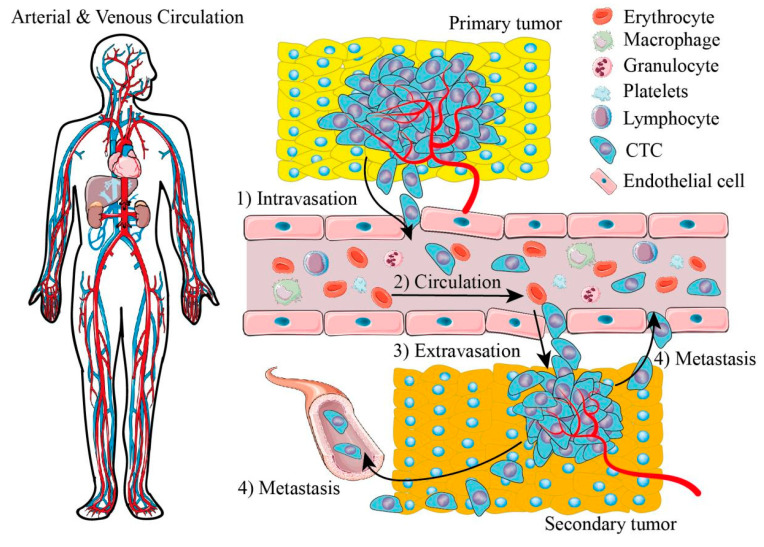
Illustrations of arterial and venous circulation and cancer metastasis. (1) Tumor cells of primary tumors invade endothelium. (2) Tumor cells circulate in blood vessels with blood cells, called circulating tumor cells (CTCs). (3) CTCs evacuate from blood vessels and invade a distant position to constitute a secondary tumor. (4) Tumor cells of secondary tumors invade blood vessels again to build another tumor site.

**Figure 2 cells-11-00905-f002:**
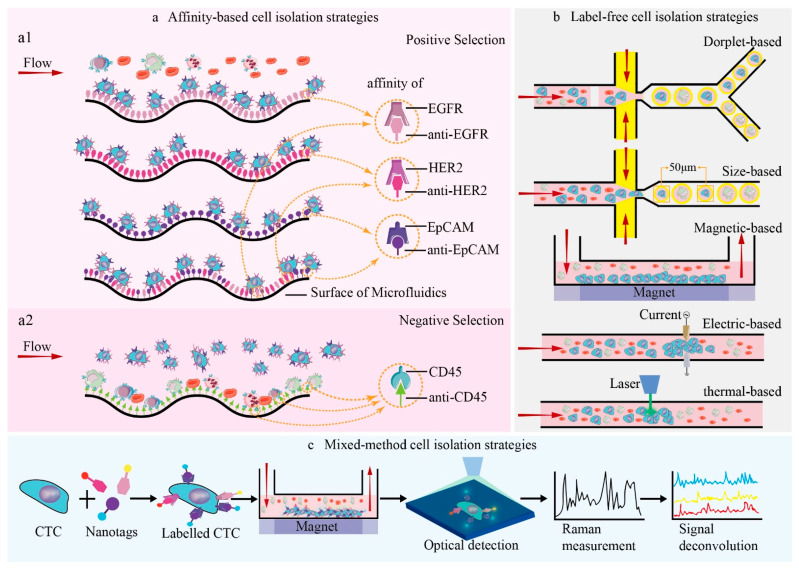
Isolation of tumor cells in microfluidic devices based on biomarkers, label-free methods, and mixed methods. (**a**) Affinity-based cell isolation, (**a1**) positive selection, (**a2**) Negative selection. (**b**) Label-free isolation strategies based on different biophysical properties. (**c**) Mixed-method cell isolation strategies based on immunomagnetic isolation and SERS.

**Figure 3 cells-11-00905-f003:**
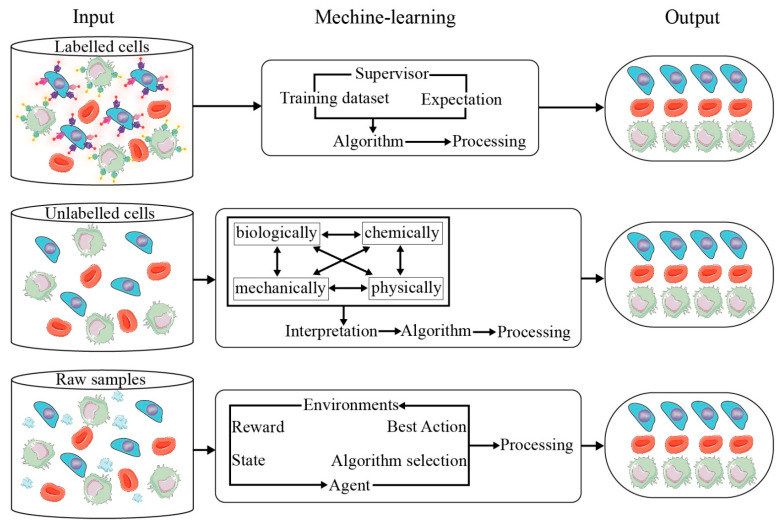
Different types of machine learning are applied in cell classification in microfluidic platforms.

## Data Availability

Not applicable.

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
