# Peer review of "Machine Learning-Driven Multiobjective Optimization: An Opportunity of Microfluidic Platforms Applied in Cancer Research"

_cells, 2022, doi:10.3390/cells11050905_

Round 1

Reviewer 1 Report

The authors tried to summarize machine learning assisted microfluidic technologies in cancer research; however, the content doesn’t match well with the title. The authors tried to cover “everything” with limited wording.

  1. Cancer research, cancer metastasis, the topic shrinks down to CTC;
  2. Many places are missing proper references, i.e. 102-105;
  3. Information of CTC capture is not up to date; many other technologies are not summarized in the text. DEP isolation technique is more than your description, which brings concerns on the literature searching skills of the authors.
  4. Figure 2 is too basic.
  5. SERS is another world. Based on the paragraphs about SERS, microfluidic SERS, the authors didn’t understand this topic well.
  6. Figure 3 is too simple. This figure doesn’t belong to scientific papers.
  7. The overall text is more like making “true statement” about many topics. The audience in this field won’t get new information from it. The logic of this manuscript doesn’t flow well. Thus, the overall value of this manuscript is limited.
  8. Last but not least, the reference style is not MDPI cells. It feels like a manuscript rejected by B&B or other journals with this kind of reference style. If so, the author should provide the revisions of previous comments. I want to see what improvement has been done. From current format, the authors not only need to revise the title, but also need to re-write the whole manuscript from “zero”.

Author Response

The authors tried to summarize machine learning assisted microfluidic technologies in cancer research; however, the content doesn’t match well with the title. The authors tried to cover “everything” with limited wording.

We appreciate the comments and suggestions made by the respected reviewer.

  1. Cancer research, cancer metastasis, the topic shrinks down to CTC;

Cancer metastasis is a key contributor of cancer incidence and fatality, thus an important topic of cancer research. CTC plays a key role in metastasis and has been widely studied in microfluidics. In most ML applications, we have added discussions for other aspects of cancer research such as droplets, cell-cell interactions, thus choose a general title. We would appreciate any specific suggestions on that. 

  1. Many places are missing proper references, i.e. 102-105;

Thank you for the pointing out this mistake. We have carefully added quotations and highlighted it in green.

  1. Information of CTC capture is not up to date; many other technologies are not summarized in the text. DEP isolation technique is more than your description, which brings concerns on the literature searching skills of the authors.

Thanks for the suggestion. We have highlighted the major and important techniques related to CTC capture and microfluidics based on ML and summarized more up to date works. We had added more discussions about DEP in cell isolation as highlighted in blue color.

  1. Figure 2 is too basic.

Thanks for suggestion. The topic of this review spans many disciplines. We feel a schematic diagram of fundamental biology involved is needed for broad readers, especially for researchers without biological background. Currently, EpCAM is a well-known antibody, but EGFR and HER2 are not very popular due to their different expression indifferent types of cancers. We also updated corresponding contents in Figure 2, including unlabeled analysis method and mixed analysis based on SERS.

  1. SERS is another world. Based on the paragraphs about SERS, microfluidic SERS, the authors didn’t understand this topic well.

We appreciate the suggestion. We rewrote this part related to SERS as highlighted in blue color.

  1. Figure 3 is too simple. This figure doesn’t belong to scientific papers.

We deleted the original figure 3 and drew a new figure for different classification of ML applications, which is more in line with the requirements of the review article.

  1. The overall text is more like making “true statement” about many topics. The audience in this field won’t get new information from it. The logic of this manuscript doesn’t flow well. Thus, the overall value of this manuscript is limited.

We appreciate this comment and have made significant efforts to revise and make the overall logic of the paper smoother.

  1. Last but not least, the reference style is not MDPI cells. It feels like a manuscript rejected by B&B or other journals with this kind of reference style. If so, the author should provide the revisions of previous comments. I want to see what improvement has been done. From current format, the authors not only need to revise the title, but also need to re-write the whole manuscript from “zero”.

Thanks for the comment. This manuscript has not been submitted to any other journal.  This is our first paper submitted to MDPI cells, thus forgot to follow the reference style of MDPI cells. The references have been updated and the whole manuscript has almost been re-written (as highlighted in red, blue, yellow colors).

Reviewer 2 Report

The paper reviews the development history of microfluidics used for cancer research and summarizes how utilization of machine learning can benefit cancer studies, particularly in biomarker detection wherein causality analysis is inspiring. It is a topic of interest to the researchers in the related areas but the paper needs very significant improvement before acceptance for publication. My detailed comments are as follows:

Following are some main comments:

 Paragraph layout needs to be rearranged. For example, describe droplet Microfluidics in detail on pages 7~8. The combination of microfluidic platforms and ML has been described in detail earlier. However, droplet Microfluidics falls under this section and should not be listed separately.

Following are some minor comments:

  1. Your manuscript needs careful editing by someone with expertise in technical English editing paying particular attention to English grammar, spelling, and sentence structure.
  2. Before and after the sentence unified. For example, the citation format is not uniform

Author Response

The paper reviews the development history of microfluidics used for cancer research and summarizes how utilization of machine learning can benefit cancer studies, particularly in biomarker detection wherein causality analysis is inspiring. It is a topic of interest to the researchers in the related areas but the paper needs very significant improvement before acceptance for publication.

We appreciate the valuable comments and suggestions made by the reviewer.

My detailed comments are as follows:

Following are some main comments:

 Paragraph layout needs to be rearranged. For example, describe droplet Microfluidics in detail on pages 7~8. The combination of microfluidic platforms and ML has been described in detail earlier. However, droplet Microfluidics falls under this section and should not be listed separately.

We thank the respected reviewer for the comments and constructive suggestion about the paragraph layout. We had made substantial changes in the paragraph layout:

  1. Moved line 122 to 131 on page 3 to line 163 on page 5 as highlighted in yellow.
  2. Moved line 267 to 299 on page 7&8 to line 206 on page 6 as highlighted in yellow.
  3. In order to make the logic clearer, we added some sentences and paragraph for systematic description and marked them in blue in the paper.

The logic of systematic description is as fellow:

We first summarized what is cancer metastasis and why is the early diagnosis important. Then we talk about the cancer-related microfluidics in general. We introduced the affinity-based and label-free based strategies of microfluidics in cell classification and mentioned causality analysis is key organizing principle for designing better microfluidic device. Next, we discussed ML algorithms applied in microfluidics. In bio-application of microfluidic ML, we mainly introduced droplet microfluidics. Then we discussed different applications of ML based on cell properties. In addition, we talk about other usage of ML in microfluidics. In the end, we conclude the potential of ML to achieve analysis of causality of different properties and disciplines.   

Following are some minor comments:

1. Your manuscript needs careful editing by someone with expertise in technical English editing paying particular attention to English grammar, spelling, and sentence structure.

Thanks for the suggestion. We have gained help from native speakers to check the grammar, spelling and sentence structure and have made substantial changes throughout the manuscripts, as highlighted in red.

2. Before and after the sentence unified. For example, the citation format is not uniform

Thanks for pointing out this mistake. We made changes to the citation to make them consistent and follow the citation format of MDPI cells.

Round 2

Reviewer 1 Report

The authors addressed most of my comments.

Reviewer 2 Report

I found that three authors put considerable effort into dealing with the referee's report. As such, the paper is greatly improved and I have no problem recommending it for publication.